# Detection and Control of *Fusarium oxysporum* from Soft Rot in *Dendrobium officinale* by Loop-Mediated Isothermal Amplification Assays

**DOI:** 10.3390/biology10111136

**Published:** 2021-11-05

**Authors:** Caiyun Xiao, Rongyu Li

**Affiliations:** 1Institute of Crop Protection, Guizhou University, Guiyang 550025, China; xcy971006@126.com; 2The Provincial Key Laboratory for Agricultural Pest Management in Mountainous Region, Guiyang 550025, China; 3College of Agriculture, Guizhou University, Guiyang 550025, China

**Keywords:** *Fusarium oxysporum*, soft rot on *Dendrobium officinale*, loop-mediated isothermal amplification, translation elongation factor-1α

## Abstract

**Simple Summary:**

In this study, we investigated the soft rot disease of *Dendrobium officinale* in Guizhou Province, and designed a rapid method to detect the pathogen is *Fusarium oxysporum* in *Dendrobium officinale* by using specific gene sequences and loop-mediated isothermal amplification technology. This method can accurately detect *Fusarium oxysporum* on soft rot plants of *Dendrobium officinale*. Therefore, the results of this study are crucial for the early diagnosis of soft rot on *Dendrobium officinale*.

**Abstract:**

Soft rot causing *Fusarium oxysporum* is one of the most destructive diseases of *Dendrobium officinale* Kimura et Migo in China that reduces *D. officinale* yield and quality. A key challenge for an integrated management strategy for this disease is the rapid and accurate detection of *F. oxysporum* on *D. officinale*. Therefore, a new loop-mediated isothermal amplification (LAMP) assay was developed for this purpose. In this study, the primers were selected and designed using the translation elongation factor-1α (*TEF-1α*) gene region as the target DNA sequence in order to screen the best system of reaction of LAMP to detect *F. oxysporum* through optimizing different conditions of the LAMP reaction, including time, temperature, concentrations of MgSO_4_, and concentrations of inner and outer primers. The optimized system was able to efficiently amplify the target gene at 62 °C for 60 min with 1.2 μM internal primers, 0.4 μM external primers, 7 mM Mg^2+^, and 5 fg/µL minimum detection concentration of DNA for *F. oxysporum*. The amplified products could be detected with the naked eye after completion of the reaction with SYBR green I. We were better able to control the effect of soft rot in *D. officinale* using fungicides following a positive test result. Additionally, the control effect of synergism combinations against soft rot was higher than 75%. Thus, LAMP assays could detect *F. oxysporum* in infected tissues of *D. officinale* and soils in field, allowing for early diagnosis of the disease.

## 1. Introduction

*Dendrobium officinale* Kimura et Migo is a perennial herb belonging to the *Orchidaceae Dendrobium* Sw. that is commonly used in traditional Chinese medicine [1,2]. Most studies have shown that *D. officinale* contains abundant mineral elements, amino acids, and fatty acids, which explains its high medicinal value [3,4]. Modern pharmacological studies have proven that its components have a variety of effects, including antitumor, enhanced immunity, and reduced blood sugar, among others [5,6]. *D. officinale* is distributed in east Asia, southeast Asia, and Australia, along with a few other countries or regions. It is mainly distributed in the southern subtropical areas of China [7,8]. However, *D. officinale* requires a very specific growth environment and particular climactic conditions. It also grows slowly so the wild resources are very scarce [9,10]. With increasing market demand for *D. officinale*, people often choose to imitate the wild cultivation of *D. officinale* [11,12]. 

The imitated wild cultivation of *D. officinale* is often planted under forests and on the shady sides of stones [13,14]. High humidity, poor ventilation, and relative sultriness are conducive to the occurrence of diseases. Nowadays, soft rot caused by *F. oxysporum* is one of the most harmful diseases in *D. officinale* production, which can occur throughout the year. In particular, soft rot is one of the diseases that frequently occurs in *D. officinale* in Guizhou province, which is mostly caused by *Fusarium oxysporum*. The incidence of soft rot is from 30 to 50% after surveying seven regions of *D. officinale* in Guizhou, according to many surveys in our lab. The mode of transmission of *F. oxysporum* is horizontal transmission, which can infect the root or root neck of the plant. The mycelium living in the soil permeates the root through the seed and then infects the stem [15,16]. The pathogens invade the tissue through the large space between the outer skin cells of the plant. The mycelium and spores enter vessels and vascular tissues and expand into all parts of the plant [17,18]. When the temperature and humidity are ideal, *F. oxysporum* is infected rapidly. Roots, stems, and leaves begin to show symptoms of disease three days after infection [19,20]. Besides the damage inflicted on tubers, *F. oxysporum* also produces mycotoxins, which are harmful to humans and animals [21,22,23]. Therefore, it is necessary to develop a rapid method for the early diagnosis of soft rot in *D. officinale* for disease control.

Currently, the traditional pathogen isolation process includes separation, purification, microscope observation, and physiological biochemical determination. Conventional PCR identification requires special equipment and experienced technicians and takes several hours to conduct. Thus, it is not suitable for application beyond the specialist level [24,25]. A more specialized detection method, loop-mediated isothermal amplification (LAMP), is a new detection method that can efficiently amplify nucleic acid. It is widely used in the rapid detection of pathogens [26,27,28]. LAMP assays consist of two pairs of primers for six regionally identified target sequences, and substitute DNA polymerase chains under constant temperature without thermal cycler for active Bst [29,30,31]. The LAMP product can be used for real-time monitoring, adding SYBR Green I, hydroxynaphthol blue (HNB), calcein, or gel electrophoresis [32,33]. Therefore, there is no need for expensive specialist equipment, such as thermal circulators, and LAMP assays are suitable for the detection and identification of pathogens [34]. *F. oxysporum* is one of the pathogens that causes soft rot in *D. officinale*, but *Fusarium.* sp. are a complex group, which is reflected in their rich genetic diversity, high genome variability, and wide range of hosts [35,36]. Therefore, we need accurate and specific molecular markers of soft rot (*F. oxysporum*) in *D. officinale*. This study is based on using the *TEF-1α* gene of *F. oxysporum* to design a new set of LAMP-specific primers, following which the conditions of the LAMP assays were explored and optimized. We use mycelium and DNA extracted from fungi on pathogenic plant tissues to test the specificity, sensitivity, and stability of this method. This work provides an early and rapid diagnostic method for soft rot (*F. oxysporum*) in *D. officinale* for proper and timely disease control.

## 2. Materials and Methods

### 2.1. Strains

*Fusarium oxysporum* strains were isolated from samples of soft rot in *D. officinale* collected from different areas of Guizhou province, China. *F. oxysporum* strains were identified by their morphological characteristics, sequence analyses of the rDNA internal transcribed spacer (ITS), and translation elongation factor 1-alpha (*TEF-1α*) genes using primers ITS4/ITS5 [37] and EF1/EF2 [38]. All strains were preserved at Guizhou University (Table 1). 

### 2.2. Isolation and Purification of Strain

The pathogens were isolated by a conventional tissue isolation method. Symptomatic tips of *D. officinale* stems and rots were collected from Guizhou province in China during 2019–2020. Samples of approximately 3–4 × 2–3 mm were taken from the margin of necrosis at the stem base of *D. officinale*, disinfected in 75% ethyl alcohol for 5 s, and then rinsed three times in sterile distilled water. Small pieces of tissue were soaked up in sterilized water with filter paper and plated on potato dextrose agar (PDA) culture at 25 °C for 7 d. The morphological characteristics of isolates, including colony texture and color, size, and conidiophores, were assessed. The purified strains were transferred to slope PDA for preservation. 

### 2.3. DNA Extraction

Mycelia of *F. oxysporum* were grown at 25 °C for 3–5 d for DNA extraction. Genomic DNA was extracted from mycelia using a fungal gDNA isolation kit (Hangzhou Biomedical Technology Co., Ltd., Hangzhou, Zhejianng, China). The concentration of DNA was determined by spectrophotometry at 260 nm and the purity of the nucleic acid was evaluated by the ratio of absorbance at 260 to 280 nm. Genomic DNA was diluted to a concentration of 50 ng/μL and then stored at −20 °C until use.

### 2.4. Primers Design

We used a molecular evolutionary genetics analysis (MEGA7.0) program to perform a phylogenetic analysis based on the *TEF-1α* gene sequence. The basic local alignment search tool (BLAST-N) software (http://blast.ncbi.nlm.nih.gov, 7 October 2021) was used to compare the sequences with other sequences of the genus *Fusarium* from the national biotechnology information center (NCBI). The alignment analysis was carried out using MEGA7.0 software. The phylogenetic tree was constructed using a data analysis model and the adjacency method (neighbor-joining). The bootstrap support value (bootstrap) was 1000. In the phylogenetic tree, the strain XY1E208 was in the same branch as *F. oxysporum*, being isolated from other *Fusarium* species (Figure 1). Therefore, the gene sequence of *TEF-1α* could be used to distinguish different species of *F. oxysporum*. We determined the base sequence of T*EF-1a* using a conserved region of *F. oxysporum* and used ClustalX software to compare and analyze the base sequence of the *TEF-1α* gene between different species of *Fusarium* and other pathogens.

LAMP primers were designed according to partial *TEF-1α* gene sequences as species-specific primers with primer explorer V5 software (http://primerexplorer.jp/e/, 13 May 2021). Six LAMP primers are shown in Figure 2 and listed in Table 2, including two external primers (F3 and B3), two internal primers (FIP and BIP), and two loop primers (F-loop and B-loop). The primers were synthesized by Sangon Biological Engineering Co., LTD (Shanghai, China) and then repackaged with ddH_2_O after being dissolved and stored at 4 °C.

### 2.5. Optimization of LAMP Reaction Conditions

LAMP reactions were conducted according to Notomi’s protocol [29]. The LAMP reactions were accomplished in 200 µL microtubes containing 2.5 µL of 10 × LAMP Master Mix (New England biolabs (Beijing) LTD., Beijing, China.), 6 mM of MgSO_4_, 1.4 mM of dNTP Mix, 0.2 µM of outer primers (F3 and B3), 1.6 µM of internal primers (FIP and BIP), 0.4 µM of loop primers, 320 U/mL Bst DNA polymerase, 1.0 µL DNA template, 3 µL of 1000 × diluted SYBR Green I (Sangon Biotech Co., Ltd., Shanghai, China), and autoclaved distilled water was used to adjust the volume to 25 µL in the Loopamp RealTime Turbidimeter LA-320C (Eiken Chemical Co., Ltd., Tokyo, Japan). The LAMP reaction conditions were optimized in terms of the concentration of Mg^2+^, outer primers, and internal primers, temperature, and time of Bst 2.0 DNA polymerase. 

For the optimization of reagents, a range of reaction temperatures (55, 58, 60, 62, 64, 65, 66, 68, and 70 °C), a range of reaction times (15, 30, 45, 60, 75, and 90 min), a range of Mg^2+^ concentrations (2, 3, 4, 5, 6, 7, 8, 9, and 10 mM), a range of inner primer concentrations (0.4, 0.8, 1.2, 1.6, and 2 µM), and a range of outer primer concentrations (0.2, 0.4, 0.6, 0.8, and 1 µM) were evaluated under otherwise identical conditions. The LAMP reactions were performed in 200 µL microfuge tubes incubated in a water bath for 60 min at 65 °C. The reactions were halted by immersion at 80 °C for 5 min, at which point reaction products were detected by fluorescent LAMP and electrophoresis, following which LAMP reactions were completed. After the LAMP reactions took place, the naked eye color changes were directly observed after adding 3 μL 1000 × SYBR Green I dye and centrifuging at the end of the tube.

### 2.6. LAMP Assays Specificity 

Eighteen fungus strains were used for the LAMP specificity test, including strains belonging to the *Fusarium* genus and other non-*Fusarium* species (Table 1). The DNA of the tested strains was extracted using a fungal genome DNA isolation kit (Hangzhou Bio-medical technology Co., Ltd. Hangzhou, Zhejiang, China). LAMP reactions were implemented with extracted DNA under optimized conditions. Each experiment was repeated three times. 

### 2.7. Detection of F. axysporum by LAMP and Conventional PCR

To evaluate the sensitivity of the LAMP assays, the DNA of *Fusarium* wilt in *D. officinale* was extracted and used as a control for LAMP amplification for specific detection. The DNA of *F. oxysporum* was diluted at a range of concentrations (10^–1^, 10^–2^, 10^–3^, 10^–4^, 10^–5^, 10^–6^, 10^–7^, 10^–8^, and 10^–9^ times) as a template for the experiment. The conventional polymerase chain reaction (PCR) mixture contains 1 µL of DNA, 1 µL of outer primers (F3/B3), 10 µL of 2 × Taq Mix DNA polymerase, 7 µL of ddH_2_O, and 20 µL of distilled autoclaved water. The reaction mixtures were incubated in a BIO-RAD TP100 PCR machine. The program was 94 °C for 5 min; 35 cycles of 94 °C for 30 s, 56 °C for 30 s, 72 °C for 1 min; 72 °C for 10 min. 

After a color reaction with fluorescent dye, the 5 μL amplification products were detected using electrophoresis with 1% agarose gel and the molecular weights of bands were determined using DNA molecular weight standards. The products were reacted for 25 min under conditions of 150V and 400A. The gels were photographed under ultraviolet light in a gel imaging system after electrophoresis. The existence of ladder bands under ultraviolet light showed that there was soft rot (*F. oxysporum*) in the *D. officinale* samples. However, no ladder bands indicated a negative result. 

### 2.8. The Feasibility Detection of LAMP Assays

To evaluate the feasibility of diagnosis by LAMP assays in the field, healthy stems were inoculated with *F. oxysporum* to simulate field-infected stems and then DNA was extracted from the stems of *D. officinale* and cultivated soil using a fungal genome DNA isolation kit (Hangzhou Biomedical Technology Co., Ltd. Hangzhou, Zhejiang, China) with apparent disease symptoms. The LAMP assays were performed as described above, and non-symptomatic stems of *D. officinale* and sterilized soil were used as controls. In addition, suspected soft rot of *D. officinale* in six areas of Guizhou province were collected, and LAMP assays could be used to detect pathogens with observations of color changes after adding SYBR Green I.

### 2.9. Prevention and Control of F. oxysporum from Soft Rot in D. officinale by LAMP Assays

Control effects of fungicide combination against *F. oxysporum* were studied in a growth chamber. The pots for planting *D. officinale* were 6.5 × 8 cm (φ × h) and filled with nutrient soils, which were inoculated with *F. oxysporum* following an acupuncture method. The samples of *D. officinale* were detected by LAMP assays before controlling soft rot with a combination of fungicides.

The fungicide combinations were diluted with 0.5% Tween-20 water to spray onto *D. officinale*. Additionally, 10 mL of fungicide (pyraclostrobin, picoxystrobin, osthole, and physcion) liquid was sprayed onto *D. officinale* after positive detection by LAMP assay. A total of 10 mL of sterile water was set as the control group. The control effects of fungicide combinations were observed and recorded after spraying for 14 d. The control effect calculation formula is DI = (∑ number of diseased leaves (stalks) × disease grade index) / (the total number of leaves (stalks) × maximum disease level) × 100. An increased value of the disease index = the DI of leaves × 0.5 + the DI of stalks × 0.5; control effect (%) = (increased value of the disease index in the processing areas—increased value of the disease index in the control areas) / increased value of the disease index in the processing areas × 100.

## 3. Results 

### 3.1. Identification of Pathogens

The disease mainly harmed stems of *D. officinale*. In the early stages of this disease, waterlogged disease spots appeared in plant stems. With the development of the disease, damaged areas expanded, leaves became yellow, stems gradually rotted from the bottom up, and finally whole plants wilted (Figure 3A). We collected samples of *D. officinale* from seven areas of the Guizhou province, isolating and purifying a total of 66 strains, which represented *XY1E208* (23 strains of *F. oxysporum*) (Appendix A). The colonies of strain *XY1E208* were circular on PDA medium with hyphae stripes. The aerial hyphae were white, flocculent, and villous. The early colonies were white, and then they changed to a pale purple pigment in the middle of the colonies (Figure 3B). The conidia of *F. oxysporum* were hyaline, ellipsoidal, unicellular, 1–2 intervals, and ranged from 4.97 to 25.58 × 1.36 to 4.33 µm in size (Figure 3C).

### 3.2. Optimization of LAMP Assays

In order to find the best LAMP reaction conditions, we carried out many LAMP reactions under different temperatures (55–70 °C), times (15–90 min), concentrations of Mg^2+^ (2–10 mM), inner primers (04–2 µM), and outer primers (0.2–1 µM). The results of SYBR Green I staining showed that the best temperature for LAMP detection of genomic DNA in *F. oxysporum* was 62 °C (Figure 4A). After testing at several reaction times, 60 min was optimal for the LAMP assays for *F. oxysporum* (Figure 4B). The agarose gel of the ladder bands showed that the reaction had high efficiency with 1.2 µM of internal primers (Figure 4C), 0.4 µM of external primers (Figure 4D), and 7 mM Mg^2+^ (Figure 4E). All LAMP reactions were carried out in a Loopamp RealTime Turbidimeter LA-320C.

### 3.3. Specificity of LAMP Assays

LAMP specificity was determined with genome DNA from *F. oxysporum* and other pathogens through direct visual observation after adding SYBR Green I stain (Table 1). After the reaction, positive samples of color turned green but the color of negative samples remained orange. LAMP primers of *F. oxysporum* based on *TEF-1α* amplification yielded positive reactions when testing for DNA, while the DNA from other pathogens were used as negative controls (Figure 5 and Appendix A).

### 3.4. Sensitivities of LAMP and Conventional PCR Assays 

The limits of the LAMP assays were evaluated with a series of concentrations (10^–1^–10^–9^ ng/μL) of *F. oxysporum* DNA template based on partial *TEF-1α* gene sequences under optimized reaction conditions. The limit of LAMP assays was 5 fg/μL for the genomic DNA of *F. oxysporum* in *D. officinale* (Figure 6A,B). However, the sensitivity of conventional PCR was every 25 μL system of *F. oxysporum* DNA purified by 5 pg/µL (Figure 6C). Therefore, the LAMP assays were more sensitive than conventional PCR assays.

### 3.5. The Feasibility Detection of LAMP Assays

All disease samples and inoculated samples of *D. officinale* were detected by LAMP assays. Our search results showed that LAMP products of positive reactions turned green after mixing with SYBR Green I while products of samples without infection remained orange (Figure 7). After observing significant symptoms, eight samples were detected to be positive of nine samples from infected tissues by LAMP assays (Figure 7). Seven samples were also positive of nine soil samples (Figure 7B). In addition, six suspected samples of soft rot in *D. officinale* collected from the Guizhou province in the field were all positive when using LAMP assays (Figure 7C). As a result, the LAMP assays could be used for detecting directly diseased samples of *F. oxysporum* in *D. officinale*, and further used for rapid diagnosis of the disease in the field.

### 3.6. Prevention and Control of Soft Rot in D. officinale by LAMP Assays

We used 21 pots of *D. officinale* plants artificially planted with an average plant height of 15 cm in this pot experiment. After pathogenic back grafting with *F. oxysporum*, 12 samples were randomly selected for LAMP testing before disease symptoms appeared. Three samples showed positive results among 12 random samples (Figure 8A). However, 10 samples showed positive results among 12 random samples after disease symptoms appeared (Figure 8B). Then, we immediately sprayed our samples with fungicides in a synergistic combination to control the soft rot in *D. officinale* plants after most samples tested positive via LAMP assays. The control effects of fungicide synergism combinations (pyraclostrobin and picoxystrobin, osthole and physcion) against soft rot were higher than 75.00% after spraying for 14 days, with control effects being 82.39 and 76.74%, respectively (Table 3). Thus, the LAMP assays can be used for detecting directly diseased samples of *F. oxysporum* in *D. officinale*.

## 4. Discussion 

The soft rot in *D. officinale* caused by *F. oxysporum* is one of the main diseases in the productive areas of China. This pathogen harms plants’ roots, causing vascular bundle disease and plant death, which can occur along the entire growth period of *D. officinale*, resulting in great losses to production [39,40,41,42]. The seriously damaged crops reported at home and abroad are eggplant, sugarcane, blackberry, cotton, etc. [43,44,45,46]. At present, chemical control is also the most common method to control plant diseases. However, the effect is often not ideal due to *F. oxysporum* being a soil-borne disease [47,48,49]. Thus, it is essential to develop an early and rapid diagnostic method for soft rot (*F. oxysporum*) in *D. officinale* for proper and timely disease control. In this study, when *F. oxysporum* primers showed positive reactions in LAMP detection assays, it could predict occurrences of the disease, which is sufficient evidence to guide further control of soft rot in *D. officinale*. *TEF-1α* is a highly conserved and ubiquitous protein, which has been widely used to study intra- and inter-species variation and phylogeny in the *Fusarium* genus [50,51]. Therefore, the *TEF-1α* region is suitable as a target for the design of LAMP primers. The results of this study also show that the designed primers were highly specific to *F. oxysporum*, thus LAMP detection could correctly distinguish *F. oxysporum* from a variety of *Fusarium* spp.

Our study showed that LAMP assays could correctly detect *F. oxysporum*. Compared with a conventional PCR assay, the LAMP assays were fast and simple because LAMP reactions are carried out under constant temperature conditions without a thermal circulator. The results of the LAMP assays can be visualized after adding SYBR Green I or Calcein without gel electrophoresis [52,53]. In our study, we used SYBR Green I dye, which showed a clear color change from orange to green to indicate a positive reaction. We also rapidly developed LAMP assays to successfully identify *F. oxysporum* in *D. officinale* with a detection limit of 5 fg/µL (Figure 5A), which was significantly higher than previous studies and more sensitive than conventional PCR tests [54,55]. The difference in detection limits by LAMP assays may be due to different sequences as targets in reactions [56]. Therefore, the optimization of a LAMP detection system is very important.

The LAMP method can detect plant pathogens based on the amplification of target DNA sequences [57,58]. We detected various specimens by extracting DNA from different types of samples, including hyphae, infected tissue of *D. officinale* and soil, and suspected samples in the field. Our search results indicated that LAMP assays could be used to directly detect *F. oxysporum* in diseased samples. These results were consistent with previous studies using LAMP assays [59,60]. The LAMP assays could detect the samples of *F. oxysporum* from plant tissues and soil, confirming that this method could be used to diagnose soft rot of *F. oxysporum* in *D. officinale*. 

Previous studies have reported that external spores have been dispersed by wind, water, people, equipment, and the movement of soil particles as they contain the fungus [61]. Therefore, it is important to prevent and control Fusarium wilt, which is maintained in environmental conditions. In this study, we have established a method to rapidly detect *F. oxysporum* that also provides early diagnosis of various diseases caused by *F. oxysporum*. Combined with results of detection by LAMP assays, we may find more accurate and effective pesticides in order to rapidly bring infection under control.

## 5. Conclusion 

In this study, a LAMP method for the detection of *F. oxysporum* was established. The LAMP assays of *F. oxysporum* were reacted for 60 min at 62 °C with 1.2 µM internal primers, 0.4 µM external primers, and 7 mM Mg^2+^, while the PCR method would usually have taken 2–6 h. Based on the *TEF-1α* gene sequence, designed primers were highly specific for *F. oxysporum*, with a detection limit of 5 fg/µL through visual inspection after SYBR Green I staining, which was significantly more sensitive than a conventional PCR test (5 pg/µL). Thus, LAMP assays can be used as an effective tool for the early diagnosis of soft rot (*F. oxysporum*) in *D. officinale* in the field for proper and timely disease control.

## Figures and Tables

**Figure 1 biology-10-01136-f001:**
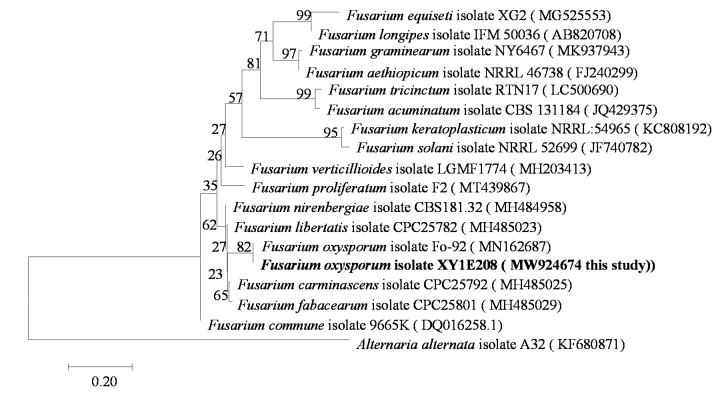
Phylogenetic tree for *F.*
*oxysporum* based on translation elongation factor-1α (*TEF-1α*) gene sequences.

**Figure 2 biology-10-01136-f002:**
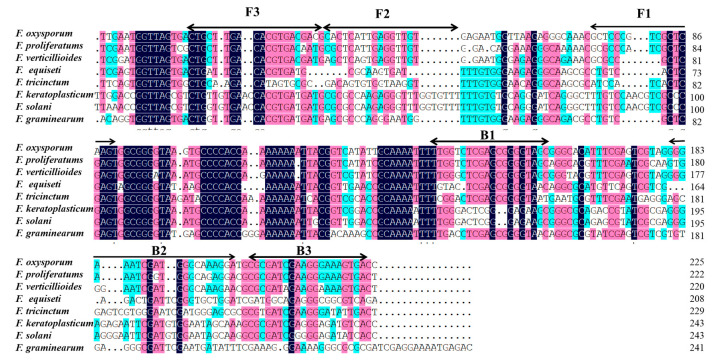
The nucleotide sequence alignment of the translation elongation factor-1α (*TEF-1α*) genes were used to design the LAMP primers.

**Figure 3 biology-10-01136-f003:**
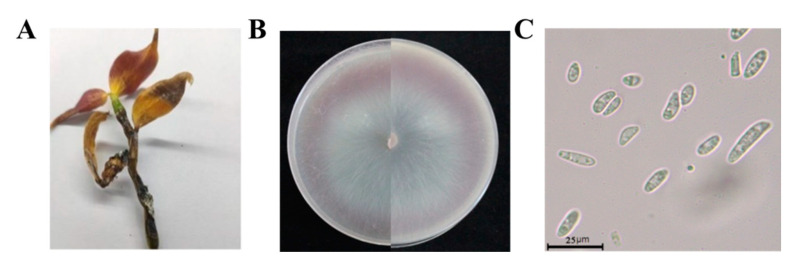
Isolation of pathogenic fungi from soft rot in *Dendrobium officinale*. (**A**) = The soft rot on *Dendrobium officinale* plant in the field; (**B**) = the morphology of strain XY1E208 after growing on PDA medium for 7 days; (**C**) = conidia morphology of strain XI1E208.

**Figure 4 biology-10-01136-f004:**
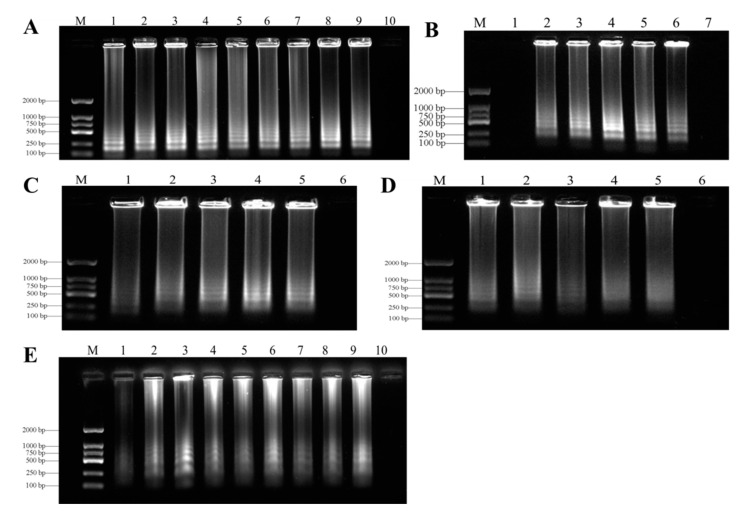
Optimization of LAMP assays. (**A**) = Optimization of LAMP reaction temperature for detection of *F.*
*oxysporum*. Lane M, Ds 2000 DNA molecular weight marker ladder; Lane 1, 55 °C; Lane 2, 58 °C; Lane 3, 60 °C; Lane 4, 62 °C; Lane 5, 64 °C; Lane 6, 65 °C; Lane 7, 66 °C; Lane 8, 68 °C; Lane 9, 70 °C; Lane 10, negative control. (**B**) = Optimization of LAMP reaction time for detection of *F.*
*oxysporum*. Lane M, Ds 2000 DNA molecular weight marker ladder; Lane 1, 15 min; Lane 2, 30 min; Lane 3, 45 min; Lane 4, 60 min; Lane 5, 75 min; Lane 6, 90 min; Lane 7, negative control. (**C**) = Optimization of LAMP reaction inner primer (FIP/BIP) concentration for detection of *F.*
*oxysporum*. Lane M, Ds 2000 DNA molecular weight marker ladder; Lane 1, 0.4 μM; Lane 2, 0.8 μM; Lane 3, 1.2 μM; Lane 4, 1.6 μM; Lane 5, 2 μM; Lane 6, negative control. (**D**) = Optimization of LAMP reaction outer primer (F3/B3) concentration for detection of *F.*
*oxysporum*. Lane M, Ds 2000 DNA molecular weight marker ladder; Lane 1, 0.2 μM; Lane 2, 0.4 μM; Lane 3, 0.6 μM; Lane 4, 0.8 μM; Lane 5, 1 μM; Lane 6, negative control. (**E**) = Optimization of LAMP reaction Mg^2+^ concentrations for detection of *F.*
*oxysporum*. Lane M, Ds 2000 DNA molecular weight marker ladder; Lane 1, 2 mM; Lane 2, 3 mM; Lane 3, 4 mM; Lane 4, 5 mM; Lane 5, 6 mM; Lane 6, 7 mM; Lane 7, 8 mM; Lane 8, 9 mM; Lane 9, 10 mM; Lane 10, negative control.

**Figure 5 biology-10-01136-f005:**
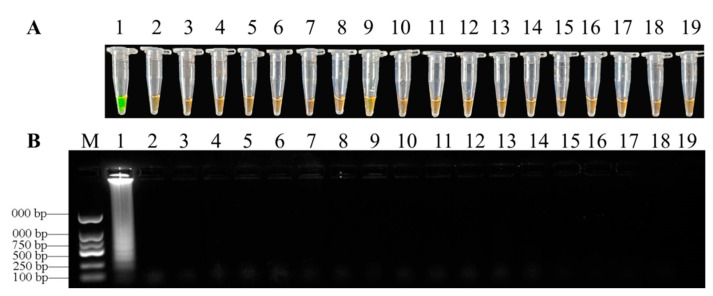
Specificity of LAMP detection of *F.*
*oxysporum*. (**A**) = Assessment based on SYBR Green I visualization of color change; (**B**) = LAMP products analyzed by agarose gel electrophoresis. 1, *Fusarium xysporum*; 2, *Fusarium*
*proliferatum*; 3, *Fusarium*
*equiseti*; 4, *Fusarium*
*solani*; 5, *Fusarium*
*chlamydosporum*; 6, *Fusarium*
*fujikuroi*; 7, *Fusarium*
*graminearum*; 8, *Colletotrichum fructicola*; 9, *Epicoccum sorghinum*; 10, *Neurospora sitophila*; 11, *Lasiodiplodia pseudotheobromae*; 12, *Trichoderma harzianum*; 13, *Botryosphaeria dothidea*; 14, *Phomopsis sp.*; 15, *Pythium ultimum*; 16, *Magnaporthe oryzae*; 17, *Rhizoctonia solani*; 18, *Botrytis cinerea*; 19, double-distilled water as negative control.

**Figure 6 biology-10-01136-f006:**
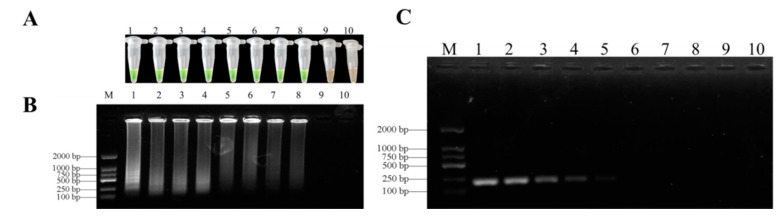
Sensitivity of LAMP and conventional PCR for detection *F.*
*oxysporum* genomic DNA. (**A**) = Detection by LAMP with SYBR Green I staining; (**B**) = LAMP products analyzed by agarose gel electrophoresis; (**C**) = conventional PCR analyzed on gel electrophoresis. The sensitivities of these assays were evaluated using serially diluted genomic DNA as follows: 1, 50 ng/µL; 2, 5 ng/µL; 3, 0.5 ng/µL; 4, 50 pg/µL; 5, 5 pg/µL; 6, 0.5 pg/µL; 7, 50 fg/µL; 8, 5 fg/µL; 9, 0.5 fg/µL; 10, negative control. M, DL 2000 DNA marker.

**Figure 7 biology-10-01136-f007:**
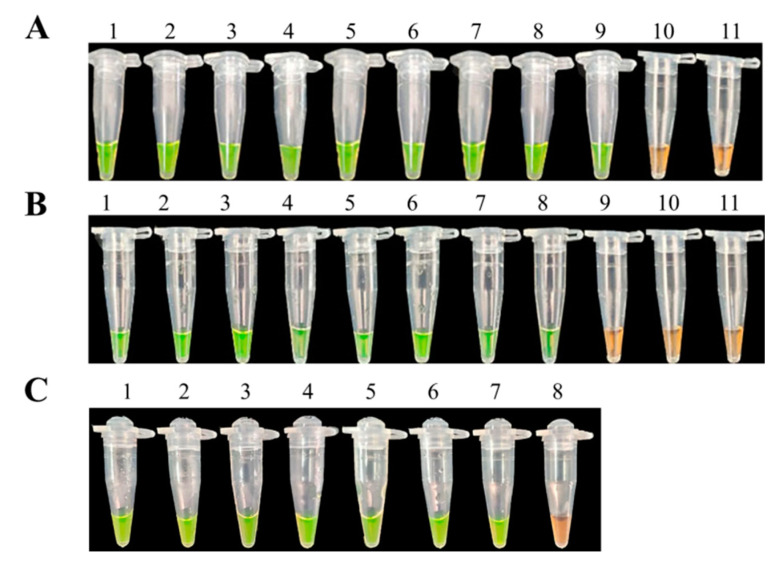
The feasibility detection of the LAMP assay. LAMP reactions were inspected by adding SYBR Green I dye. (**A**) = LAMP detection of *F.*
*oxysporum* from tissues after infection: 1, purified genomic DNA of *F.*
*oxysporum* (positive control); 2–10, DNA from *D. officinale* stems infected by *F.*
*oxysporum*; 11, DNA from healthy *D. officinale* stems (negative control); (**B**) = LAMP detection of *F.*
*oxysporum* from soil after infection: 1, purified genomic DNA of *F.*
*oxysporum* (positive control); 2–10, DNA extracted from soil after infection; 11, DNA from sterilized soil (negative control); (**C**) = LAMP detection of field issues: 1, purified genomic DNA of *F.*
*oxysporum* (positive control); 2–7, DNA extracted from field issues; 8, DNA from healthy *D. officinale* stems (negative control).

**Figure 8 biology-10-01136-f008:**
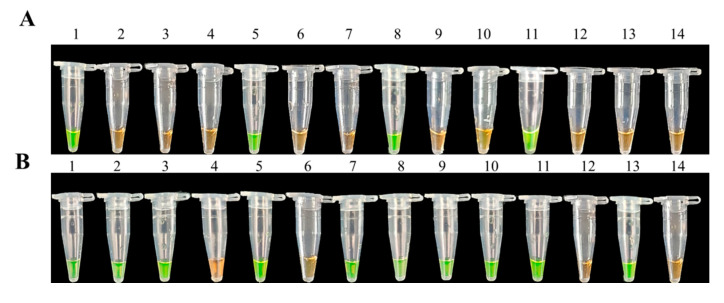
LAMP assay random sampling detection of *F.*
*oxysporum* in potted plants of *D. officinale* before pesticide control. (**A**) = LAMP assay to detect *F.*
*oxysporum* in potted plants of *D. officinale* from random samples before the symptoms of soft rot were exposed in pots: 1, purified genomic DNA of *F.*
*oxysporum* (positive control); 2–13, DNA from *D. officinale* stems before the symptoms of soft rot were exposed in pots; 11, DNA from healthy *D. officinale* stems (negative control); (**B**) = LAMP assay to detect *F.*
*oxysporum* in potted plants of *D. officinale* from random samples after the symptoms of soft rot were exposed in pots: 1, purified genomic DNA of *F.*
*oxysporum* (positive control); 2–13, DNA from *D. officinale* stems after the symptoms of soft rot were exposed in pots; 11, DNA from healthy *D. officinale* stems (negative control).

**Table 1 biology-10-01136-t001:** Different strains of fungus used in this study to test the specificity of the LAMP assay.

Species	Host	Geographical Location	Number of Strains	LAMP detection	
				Agarose Gel	SYBR Green I
** *Fusarium oxysporum* **	** *Dendrobium officinale* **	**Xingyi**, **Guizhou**	1	+	+
*Fusarium proliferatum*	*Dendrobium officinale*	Huaxi, Guiyang	1	−	−
*Fusarium equiseti*	*Dendrobium officinale*	Huaxi, Guiyang	1	−	−
*Fusarium solani*	*Dendrobium officinale*	Libo, Guiyang	1	−	−
*Fusarium chlamydosporum*	*Dendrobium officinale*	Anlong, Guiyang	1	−	−
*Fusarium fujikuroi*	Plum	Huishui, Guiyang	1	−	−
*Fusarium graminearum*	Kiwi Fruit	Xifeng, Guiyang	1	−	−
*Colletotrichum fructicola*	*Dendrobium officinale*	Sansui, Guizhou	1	−	−
*Epicoccum sorghinum*	*Dendrobium officinale*	Xingyi, Guizhou	1	−	−
*Neurospora sitophila*	*Dendrobium officinale*	Jinping, Guizhou	1	−	−
*Lasiodiplodia pseudotheobromae*	*Dendrobium officinale*	Jinping, Guizhou	1	−	−
*Trichoderma harzianum*	Soil	Jinping, Guizhou	1	−	−
*Botryosphaeria dothidea*	*Dendrobium officinale*	Jinping, Guizhou	1	−	−
*Phomopsis* sp.	Kiwi Fruit	Xifeng, Guizhou	1	−	−
*Pythium ultimum*	*Dendrobium officinale*	Huaxi, Guizhou	1	−	−
*Magnaporthe grisea*	*Oryza sativa*	Huaxi, Guizhou	1	−	−
*Rhizoctonia solani*	*Oryza sativa*	Huaxi, Guizhou	1	−	−
*Botrytis cinerea*	Kiwi Fruit	Xifeng, Guizhou	1	−	−

*Note:* bold letter = the *Fusarium oxysporum* used for specific detection in this study.

**Table 2 biology-10-01136-t002:** Primers used for LAMP assays to detect *F.*
*oxysporum*.

Primer Name	Sequence (5′–3′)	Length
F3	ACTGCTTGACACGTGACG	18
B3	CACTTTCCCTTCGATCGCG	19
FIP	ACTTACCCCGCCACTTGAGCACGCACTCATTGAGGTTGTG	40
BIP	TTGGTCTCGAGCGGGGTAGCTCCTTTGCCCATCGATTTCC	40
LF	CGTTTGCCCTCTTAACCATTCT	22
LB	GGGCACATTTCGAGTCGTAGG	21

**Table 3 biology-10-01136-t003:** Effect of fungicide synergism combinations against soft rot in *D. officinale*.

Treatment	Concentration (g a.i./hm^2^)	Average Disease Index before Spraying Fungicide	Average Disease Index 14 Days after Spraying Fungicide	Control Efficacy (%)
CK	/	32.08	87.41	/
Pyraclostrobin(A)	4.9635	35.83	44.44	84.25 a
Picoxystrobin(B)	11.0865	43.47	55.65	78.07 c
A:B(1:3)	4.128	47.78	57.5	82.39 ab
Osthole(C)	16.839	41.59	53.06	79.29 bc
Physcion(D)	192.3	35.65	54.21	66.47 d
C:D(7:1)	18.195	38.33	51.11	76.74 c

Note: Different lowercase letters in the same column show significant difference at 0.05 level by Duncan’s multiple range test.

## Data Availability

Sequencing data are available from https://www.ncbi.nlm.nih.gov/nuccore/MW924674.

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
