# Peer review of "Detection and Control of Fusarium oxysporum from Soft Rot in Dendrobium officinale by Loop-Mediated Isothermal Amplification Assays"

_biology, 2021, doi:10.3390/biology10111136_

Round 1
Reviewer 1 Report
Research paper Detection and control of Fusarium oxysporum from soft rot of Dendrobium officinale by loop-mediated isothermal amplification assays presents novel technique that can be applied for early detection of a mould causing disease of a crop. As this is very significant research ares, especially due to growing problem with a food availability and the spreading fo diseases abroda, I highly wellcome this initiative. However this paper has some issues needed to be adresses before publishing.First of all the paper needs to be more precisely written, without duplication and narration. For example see line 22-24 in Abstract - long sentence duplicating already written! Sentence 25-27 is replication of a second sentence. And this just goes on.
Major concerns:
1. As you stated - tissue was "disinfected in 75% ethyl alcohol for 5 s and rinsed three times in sterile distilled water. Small pieces of tissue were sterilized". How you obtained microbes from a sterilized tisse as sterilisation, by its definition, is a proces where all microbes are destroyed!
2. Correct the line 181 (5 ml amplification product!!)
3. Line 192 you have a dot with n meaning
4. "For sake of evaluate" - To evaluate
5. Fungicides are not toxic to mould (line 199)!
6. Did you introduced validated strain to your research? I highly recommend to do so in fungicide testing especialy. On the other hand, you connected two different research in one paper with no visible connected of those. Perhaps you can reconsider to omit part of fungicide activity as these are not adding value to the testing of LAMP system as a possible beneficial method.
Author Response
Reviewer #1:
Comment 1: First of all the paper needs to be more precisely written, without duplication and narration. For example see line 22-24 in Abstract - long sentence duplicating already written! Sentence 25-27 is replication of a second sentence. And this just goes on.
Response: Thanks very much for your attention to our paper! We had revised all details in the revised manuscript, including abstract,words, grammar and presentation errors.
Comment 2: As you stated - tissue was "disinfected in 75% ethyl alcohol for 5 s and rinsed three times in sterile distilled water. Small pieces of tissue were sterilized". How you obtained microbes from a sterilized tisse as sterilisation, by its definition, is a proces where all microbes are destroyed!
Response: Thank you very much for valuable comments of reviewer. This part was our narrative error, and for this part of the error, we had modified it. The specific modification was: “The tissue of approximately 3-4 mm × 2-3 mm were taken from the margin of necrosis at the stem base of D. officinale, and disinfected in 75% ethyl alcohol for 5 s, and then rinsed three times in sterile distilled water. Small pieces of tissue were soaked up steriliaed water with filter paper and plated on potato dextrose agar (PDA) culturing at 25℃ for 7 d.”
Comment 3: Correct the line 181 (5 ml amplification product!!)
Response: Thank you very much for your detailed comments of reviewer! We had corrected “5 mL amplification product” to “After color reaction with fluorescent dye, the 5 mL amplification products were detected using electrophoresis with 1% agarose gel, and the molecular weight of bands were determined by DNA molecular weight standard.”
Comment 4: Line 192 you have a dot with n meaning
Response: Thank you very much for your detailed comments of reviewer! This dot was a written form of the biological company to which the fungal gDNA isolation kit used to extract fungal DNA in the article belongs.
Comment 5: "For sake of evaluate" - To evaluate
Response: Thank you very much for your detailed comments of reviewer! We had corrected “For sake of evaluate” to “To evaluate the feasibility for diagnosis by LAMP assays in field, the healthy stems had been inoculated with F. oxysporum to simulate field infected stems, and then DNA were extracted from stems of D.officinale and cultivated soil by using fungal genome DNA isolation kit (Hangzhou bio-bedical technology Co., Ltd.).”
Comment 6: Fungicides are not toxic to mould (line 199)!
Response: Thank you very much for your detailed comments of reviewer! The pathogen in this article was Fusarium oxysporum. In our study, based on the LAMP test results, we used fungicides to control the plants before the obvious disease symptoms appeared. The results showed that fungicides had a good control effect on Fusarium oxysporum in time.
Comment 7: Did you introduced validated strain to your research? We highly recommend to do so in fungicide testing especially. On the other hand, you connected two different research in one paper with no visible connected of those. Perhaps you can reconsider to omit part of fungicide activity as these are not adding value to the testing of LAMP system as a possible beneficial method.
Response: Thank you very much for your sincere suggestions on the structure of the article content of reviewer! The strain XY1E208 in this article was a fungus identified as Fusarium oxysporum through morphological and molecular methods. We also used this strain in the fungicide experiment. The different gene registration numbers of this strains were MW843454, MW924673, MZ054640, MW924674, MZ615701, respectively. Besides, in view of the problem that there was no obvious connection between LAMP assay and fungicide activity, we had taken your reliable opinion and omitted the content of the fungicide activity in our article.
Reviewer 2 Report
This is an example of using a well established type of method (LAMP assay) for the identification of an important plant-pathogenic fungus. Despite the novelty is medium, these results can be of interest, as they present an in-depth optimization of the method for a specific goal.
However, the reliability of the method, mainly in terms of potential false negatives, needs to be more clearly substantiated. How many strains of Fusarium oxysporum were taken into account? Figure 1 shows two strains of this organism. In this context, the following phrase (l. 137) is unclear: "isolation and purification of total 66 strains, which represented 237 by numbers for XY1E208(23 strains of F. oxysporum)". Please clarify the number of strains used to develop the target assays.
The second part of the study addressed a different topic and the use of the aforementioned LAMP assay in this part is very limited. I suggest to focus on the first part (assay development) only.
Other factual revisions are limited but the text needs a very thorough language revision in terms of grammar.
Section 2.2.: The title is not quite appropriate as the method is obviously not selective for F. oxysporum.
In this context, the 1st sentence in section 2.3. (Mycelia of F. oxysporum grew ...) is also not quite suitable.
Table 3 - please explain the meaning of symbols used in the equation, and the impact of the equation itself.
The scope and impact of Discussion is limited, there is a number of repeats from previous sections.
Author Response
Comment 1: However, the reliability of the method, mainly in terms of potential false negatives, needs to be more clearly substantiated. How many strains of Fusarium oxysporum were taken into account? Figure 1 shows two strains of this organism. In this context, the following phrase (l. 137) is unclear: "isolation and purification of total 66 strains, which represented 237 by numbers for XY1E208 (23 strains of F. oxysporum)". Please clarify the number of strains used to develop the target assays.
Response: Thank you for your accurate questions on LMAP detection technology in this paper! In our study, 23 strains of Fusarium oxysporum were identified from 66 strains (Supplementary Fig. S1). However, in this article, we only selected one strain, XY1E208, as the representative of these 23 strains of Fusarium oxysporum, and did more gene sequencing and identification (The different gene registration numbers of this strains were MW843454, MW924673, MZ054640, MW924674, MZ615701, respectively, as well as subsequent LAMP experiments and fungicide experiments. In addition, the LAMP test detection of these 23 strains had been also completed, and all of them were positive (Supplementary Fig. S2). Moreover, all of LAMP reactions were carried out in the Loopamp RealTime Turbidimeter LA-320C (Eiken Chemical Co., Ltd., Tokyo, Japan). This instrument is specially used for LAMP assay, which can be observed turbidity change of product in real time to avoid false positives.
Comment 2: The second part of the study addressed a different topic and the use of the aforementioned LAMP assay in this part is very limited. I suggest to focus on the first part (assay development) only.
Response: The reviewer's remark about LAMP assay and fungicide activity was constructive. Thanks very much for your attention to our paper! We were also aware that the content of these two parts was not highly relevant. We omitted some of the fungicide active content, and left the content of the control effect (Table 4). Because this part was detected Fusarium oxysporum by LAMP assay when the plants had no obvious disease symptoms. Then we had better effect of using fungicides in advance to prevent and control diseases, which was reflected the importance of early diagnosis of LAMP for the follow-up prevention and control of diseases.
Comment 3: Other factual revisions are limited but the text needs a very thorough language revision in terms of grammar.
Response: Thanks very much for your attention to our paper! We had revised all details in the revised manuscript, including words, grammar and presentation errors.
Comment 4:Section 2.2.: The title is not quite appropriate as the method is obviously not selective for F. oxysporum.
Response: Thanks very much for your detailed comments to our paper! We had corrected to “2.2. Isolation and purification of strain”.
Comment 5: In this context, the 1st sentence in section 2.3. (Mycelia of F. oxysporum grew ...) is also not quite suitable.
Response: Thanks very much for your detailed comments to our paper! We had corrected to “Mycelia of F. oxysporum were grew at 25℃ for 3-5 d for DNA extraction. Genomic DNA was extracted from mycelia using fungal gDNA isolation kit (Hangzhou bio-bedical technology Co., Ltd.).”
Comment 6: Table 3 - please explain the meaning of symbols used in the equation, and the impact of the equation itself.
Response: Thanks very much for your detailed comments to our paper! Considering that there was no obvious correlation between the two parts of LAMP assay and fungicides activity, we had deleted part of the fungicide activity content, including the content in Table 3, so the data in Table 3 was not be explained.
Comment 7: The scope and impact of Discussion is limited, there is a number of repeats from previous sections.
Response: Thanks very much for your rigorous comments in our paper! Regarding the content of the discussion part, we had made detailed corrections.
The following was a detailed explanation:
The soft rot in D.officinale caused by F. oxysporum is one of mainly diseases in producing areas of China. The pathogens harm plant from root, causing vascular bundle disease and plant death, which can occur in the whole growth period of D.officinale, resulting in great losses to production [39,40,41,42]. The seriously damaged crops reported at home and abroad are eggplant, sugar cane, blackberry , cotton, and so on [43,44,45,46]. At present, chemical control is also the most common method to control plant diseases. But the effect is often not ideal, due to F. oxysporum as a soil-borne disease [47,48,49]. Thus, it is essential to develop an early and rapid diagnostic method a of soft rot (F. oxysporum) in D.officinale for proper disease control in time. In this study, when F. oxysporum primers showed positive reaction in LAMP detection, it could predict occurrence of disease, which is sufficient evidence to guide further control of soft rot in D.officinale. TEF-1α is a highly conserved and ubiquitous protein, which has been widely used to study intra and inter species variation and phylogeny for Fusarium genus [50,51]. Therefore, TEF-1α region as target is suitable for design of LAMP primers. The results of this study also showed that designed primers were highly specific to F. oxysporum, thus LAMP detection could correctly distinguish F. oxysporum from a variety of Fusarium spp.
Our study showed that LAMP assays could be correctly detected F. oxysporum. Compared with conventional PCR, LAMP assays were fast and uncomplicated. Because LAMP reactions are carried out under constant temperature conditions without a thermal circulator. The results of LAMP can be visualized after adding SYBR Green I or Calcein without gel electrophoresis [52,53]. In our study, we used SYBR Green I dye, which showed a clear color change from orange to green to indicate a positive reaction. We also developed rapidly LAMP assays to successfully identify F. oxysporum in D.officinale with detection limit of 5 fg/µL (Fig. 5A), which was significantly higher than previous studies, and sensitive than conventional PCR [54,55]. The difference of detection limits by LAMP assays may be due to different sequences as targets in reactions [56]. Therefore, the optimization of system of LAMP reaction is very important.
LAMP method can detect plant pathogens basing on amplification of target DNA sequence [57,58]. We detected various specimens extracting DNA from different types of samples, including hyphae, infected tissue of D.officinale and soil, and suspected samples in field. Search results indicated that LAMP assays could be used to detect directly F. oxysporum from diseased samples. These results were consistent with previous studies by LAMP assay[59,60]. The LAMP assays could be detected the sampls of F. oxysporum from plant tissues and soil confirming that this method could be used to inchoate diagnose soft rot of F. oxysporum in D.officinale and control in time.
Previous studies have reported that external spores in the parts have been dispersed by wind, water, people and equipment, and the movement of soil particles they contain the fungus[61]. Therefore, it is important to pevention and control Fusarium wilt, which are remained in environmental conditions. In this study, we have established a method to rapidly detect F. oxysporum, and it also provided early diagnosis of various diseases caused by F. oxysporum. Combined with results of detection by LAMP assays, people could accurately find more effective pesticides in order to achieve a rapid control effect, which also provides a more accurate basis for choice of pesticides category.

Reviewer 3 Report
Your work is interested, written well, and organized.
This manuscript reports on a study of Detection and control of Fusarium oxysporum from soft rot of Dendrobium officinale by loop-mediated isothermal amplification assays. The study design meets the general standards and from what I can judge the data is being collected and analyzed appropriately. This work is an unpublished manuscript with relevant information that should be made public in a scientific journal for discussion among scientists working in the field.
However, there are some comments should be considered before publishing, in this way, the social and scientific relevance of the manuscript would be improved:
As it is a section of Biochemistry and Molecular Biology in the journal, the authors should incorporate the importance of the detection and control of diseases caused by Fusarium oxysporum or the biotic stress, which represents the result of the damage caused to an organism such as cape gooseberry by other living organisms such as Fusarium, considering that all scientific advances in relation to Fusarium It will open a range of technological opportunities for the integrated management of lethal diseases caused by Fusarium, as is the case of Tropical Race 4 in bananas, recently detected in Colombia and Peru, and for traditional crops like Cape gooseberry.
Discussion
I continue to add a paragraph that summarizes the importance, usefulness and social relevance, contemporary of the study, specifically pointing out the Impact, Benefit and Social Projection, something like this:
Studies such as those of Olivares et al. [61] reveal that external spores in the parts that remain above ground level (field conditions) are dispersed by wind, water, people and equipment, and by the movement of soil particles they contain the fungus, hence the importance and benefit of considering the presence of Fusarium wilt in crops of commercial importance and favorable environmental conditions such as soil, climate, management.
The results reported here could be of practical and scientific use for future lines of research by pioneers in Plant Protection sciences and biotic interactions in developing countries such as Colombia, Peru and Venezuela, where the economic losses caused by Fusarium diseases are significant.
References
I suggest the authors adding recent references which address the issue in question of Fusarium in commercial crops. Suggested citations are for genuine scientific reasons that emphasize the current topic of study in context
- Olivares, B.O.; Rey, J.C.; Lobo, D.; Navas-Cortés, J.A.; Gómez, J.A. & Landa, B.B. Fusarium Wilt of Bananas: A Review of Agro-Environmental Factors in the Venezuelan Production System Affecting Its Development. Agronomy.2021, 11, 986. https://doi.org/10.3390/agronomy11050986
Author Response
Comment 1: I had continued to add a paragraph that summarizes the importance, usefulness and social relevance, contemporary of the study, especifically, pointing out the impact, benefit and social projection, something like this.
Studies such as those of Olivares et al. [61] reveal that external spores in the parts that remain above ground level (field conditions) are dispersed by wind, water, people and equipment, and by the movement of soil particles they contain the fungus, hence the importance and benefit of considering the presence of Fusarium wilt in crops of commercial importance and favorable environmental conditions such as soil, climate, management.
Response: Thank you so much for your considered comments to our paper! We had written this part “Previous studies have reported that external spores in the parts have been dispersed by wind, water, people and equipment, and the movement of soil particles they contain the fungus[61]. Therefore, it is important to pevention and control Fusarium wilt, which are remained in environmental conditions. In this study, we have established a method to rapidly detect F. oxysporum, and it also provided early diagnosis of various diseases caused by F. oxysporum. Combined with results of detection by LAMP assays, people could accurately find more effective pesticides in order to achieve a rapid control effect, which also provides a more accurate basis for choice of pesticides category.”
[61]. Olivares, B.O.; Rey, J.C.; Lobo, D.; Navas-Cortés, J.A.; Gómez, J.A. & Landa, B.B. Fusarium Wilt of Bananas: A Review of Agro-Environmental Factors in the Venezuelan Production System Affecting Its Development. Agronomy. 2021, 11, 986. https://doi.org/10.3390/agronomy11050986.
Comment 2: I suggest the authors adding recent references which address the issue in question of Fusarium in commercial crops. Suggested citations are for genuine scientific reasons that emphasize the current topic of study in context.
Response: Thank you so much for your thoughtful comment! We had revised the relevant references.
- Li, H.S., Ye, W., Wang, Y., Chen, X.H. RNA sequencing-based exploration of the effects of far-red light on lncRNAs involved in the shade-avoidance response of D. officinale. PeerJ, 2021, e10769, doi: 10.7717/peerj.10769
- Zhang, M.Z., Yu, Z.M., Zeng, D.Q., Si,C. Transcriptome and Metabolome Reveal Salt-Stress Responses of Leaf Tissues from Dendrobium officinale. Biomolecules, 2021, 736, doi: 10.3390/biom11050736.
- Zhang ,J., Li, T., Cai, Y.F., Wang, Y.Z. Genetic and environmental effects on allometry of the medicinal plant Dendrobium officinale (Orchidaceae) from Yunnan, southwest China. Pakistan Journal of Botany, 2021, 53(5), doi: 10.30848/PJB2021-5(14).
Round 2
Reviewer 1 Report
Dear authors, although you rewrote and corrected some parts of the paper there are still some large gaps regarding language. Thus, the paper is hard to follow and some awkward sentences can be found through the paper. I strongly suggest to find native English speaker with some scientific background to correct this paper.
For example
"and it also provided early diagnosis of various diseases
caused by F. oxysporum" - provides
"which also provides a more accurate basis for choice of pesticides category"
"while PCR method usually taked 2-6 h" - take is irregular verb!
In the conclusion section you left part regarding pesticides although you removed these results!
"Combined with early diagnostic results of soft rot on in D.officinale by LAMP assays and synergistic combinations of fungicides, this study had achieved excellent control effects of soft rot on in D.officinale."
Author Response
November 2, 2021
Reviewer #1:
Comment 1: Dear authors, although you rewrote and corrected some parts of the paper there are still some large gaps regarding language. Thus, the paper is hard to follow and some awkward sentences can be found through the paper. I strongly suggest to find native English speaker with some scientific background to correct this paper.
For example "and it also provided early diagnosis of various diseases
caused by F. oxysporum" – provides "which also provides a more accurate basis for choice of pesticides category" "while PCR method usually taked 2-6 h" - take is irregular verb!
Response: Thanks very much for your attention to our paper! We had found native English speaker with some scientific background to correct this paper. To revised all details in the revised manuscript, including abstract,words, grammar and presentation errors.
For example:
"and it also provided early diagnosis of various diseases caused by F. oxysporum". We have corrected it to “In this study, we have established a method to rapidly detect F. oxysporum, that also provides early diagnosis of various diseases caused by F. oxysporum.”
"while PCR method usually taked 2-6 h". We have corrected it to “In this study, a LAMP method for the detection of F. oxysporum was established. The LAMP assays of F. oxysporum were reacted for 60 min at 62°C with 1.2 µM internal primers, 0.4 µM external primers, and 7 mM Mg2+, while the PCR method would usually have taken 2–6 h.”
Comment 2: In the conclusion section you left part regarding pesticides although you removed these results!
"Combined with early diagnostic results of soft rot on in D.officinale by LAMP assays and synergistic combinations of fungicides, this study had achieved excellent control effects of soft rot on in D.officinale."
Response: Thank you very much for your detailed comments of this paper. I have deleted the description of this part. "Combined with early diagnostic results of soft rot on in D.officinale by LAMP assays and synergistic combinations of fungicides, this study had achieved excellent control effects of soft rot on in D.officinale." We have corrected the Conclusion to “In this study, a LAMP method for the detection of F. oxysporum was established. The LAMP assays of F. oxysporum were reacted for 60 min at 62°C with 1.2 µM internal primers, 0.4 µM external primers, and 7 mM Mg2+, while the PCR method would usually have taken 2–6 h. Based on the TEF-1α gene sequence, designed primers were highly specific for F. oxysporum, with a detection limit of 5 fg/µL through visual inspection after SYBR Green I staining, which was significantly more sensitive than a conventional PCR test (5 pg/µL). Thus, LAMP assays can be used as an effective tool for the early diagnosis of soft rot (F. oxysporum) in D. officinale in the field for proper and timely disease control.”